# MODEL TELLS YOU WHAT TO DISCARD: ADAPTIVE KV CACHE COMPRESSION FOR LLMS

**Suyu Ge**[1]*, **Yunan Zhang**[1]*, **Liyuan Liu**[2]*, **Minjia Zhang**[2], **Jiawei Han**[1], **Jianfeng Gao**[2]
[1]University of Illinois Urbana-Champaign, [2]Microsoft
{suyuge2,yunanz2,hanj}@illinois.edu
{lucliu,minjiaz,jfgao}@microsoft.com

## ABSTRACT

In this study, we introduce adaptive KV cache compression, a plug-and-play method that reduces the memory footprint of generative inference for Large Language Models (LLMs). Different from the conventional KV cache that retains key and value vectors for all context tokens, we conduct targeted profiling to discern the intrinsic structure of attention modules. Based on the recognized structure, we propose FastGen, which constructs the KV cache in an adaptive manner: evicting long-range contexts on attention heads emphasizing local contexts, discarding non-special tokens on attention heads centered on special tokens, and only employing the standard KV cache for attention heads that broadly attend to all tokens. Moreover, with the lightweight attention profiling used to guide the construction of the adaptive KV cache, FastGen can be deployed without resource-intensive fine-tuning or re-training. In our experiments across various asks, FastGen demonstrates substantial reduction on GPU memory consumption with negligible generation quality loss.

## 1 INTRODUCTION

Based on the Transformer architecture, autoregressive language models have attracted extensive attention (OpenAI, 2023; Touvron et al., 2023b). Along with the increase of model size, these models present significant challenges in terms of computational complexity and GPU memory consumption (Shazeer et al., 2017). Since these models achieve remarkable success across diverse applications, there is a pressing need for serving these models in an economically feasible manner.

The generative inference of LLMs usually involves using the *KV Cache* mechanism to improve the generation speed. KV cache stores previously computed Key/Value vectors in attention calculation and reuses those values for the current token generation. As such, it avoids recalculations of previous tokens at each token generation step at the cost of extra memory consumption. Despite being a prominent technique, the memory consumption of KV cache increases rapidly as the model size and generation length increase, drastically increasing the pressure of on-device memory.

When memory usage exceeds GPU capacity, the generative inference of LLMs typically resort to offloading (Aminabadi et al., 2022; Sheng et al., 2023). While these methods help mitigate the pressure on the scarce GPU memory from using KV cache, offloading KV cache to CPU/NVMe can still add non-trivial overhead to generative inference performance due to the limited PCIe bandwidth between the GPU and CPU on many devices. Therefore, it becomes a crucial task to reduce the memory footprint of KV cache without costly retraining or fine-tuning.

Our study starts from the observation (Figure 1) that there are abundant structures observed in attention modules (Michel et al., 2019; Voita et al., 2019; Clark et al., 2019; Wang et al., 2020; Child et al., 2019), and not all attention modules need to attend to all tokens (Liu et al., 2023b; Zhang et al., 2023; Liu et al., 2023a). Intuitively, harvesting such structures and compressing cached vectors could substantially reduce memory consumption and accelerate text generation.

Based on this intuition, we propose FastGen to *accelerate the generative inference by adaptively compressing the KV cache on the fly*. First, we employ an efficient profiling algorithm to recognize

---
*Authors contributed equally to this research. Code is available is at https://github.com/machilusZ/FastGen

the structural patterns for attention modules. Under the guidance of this profiling, we then construct the KV cache for various modules adaptively. With this diagnose-before-compress approach, FastGen effectively reduces the memory footprint of KV cache while preserving the model quality.

---

**Algorithm 1:** FastGen–Prompt Encoding.

**Input:** Feasible Policy Set ($\mathcal{C}$), Prompt
**Output:** Adaptive KV Cache
1 **for** *Attention Head $H_i$ in LLM* **do**
2 $\quad \boldsymbol{K}^i, \boldsymbol{Q}^i, \boldsymbol{V}^i \leftarrow H_i(\text{Prompt})$
3 $\quad \boldsymbol{A}^i \leftarrow \text{softmax}(\boldsymbol{Q}^i \boldsymbol{K}^{i^T})$
4 $\quad \boldsymbol{C}^i \leftarrow \text{apply Equation 1 to } \boldsymbol{A}^i$
$\quad$ /* $\text{C}^i$: optimal policy */
5 $\quad \boldsymbol{K}^i_{\boldsymbol{C}^i}, \boldsymbol{V}^i_{\boldsymbol{C}^i} \leftarrow f(\boldsymbol{K}^i, \boldsymbol{V}^i, \boldsymbol{C}^i)$
6 $\quad \hat{\boldsymbol{K}}^i, \hat{\boldsymbol{V}}^i \leftarrow \boldsymbol{K}^i_{\boldsymbol{C}^i} \boldsymbol{V}^i_{\boldsymbol{C}^i}$
7 **return** $\{\boldsymbol{C}^i, \hat{\boldsymbol{K}}^i, \hat{\boldsymbol{V}}^i\}$

**Algorithm 2:** FastGen–Token Generation.

**Input:** Adaptive KV cache ($\{\boldsymbol{C}^i, \hat{\boldsymbol{K}}^i, \hat{\boldsymbol{V}}^i\}$)
**Output:** Generated Text
1 $\boldsymbol{z}_0 \leftarrow$ last prompt token
2 **for** $j \in \{1, \cdots, \textit{Max Generate Length}\}$ **do**
3 $\quad$ **for** *Attention Head $H_i$ in LLM* **do**
4 $\quad\quad \boldsymbol{K}^i, \boldsymbol{Q}^i, \boldsymbol{V}^i \leftarrow H_i(\boldsymbol{z}_{j-1}, \hat{\boldsymbol{K}}^i, \hat{\boldsymbol{V}}^i)$
5 $\quad\quad \boldsymbol{K}^i_{\boldsymbol{C}^i}, \boldsymbol{V}^i_{\boldsymbol{C}^i} \leftarrow f(\boldsymbol{K}^i, \boldsymbol{V}^i, \boldsymbol{C}^i)$
6 $\quad\quad \hat{\boldsymbol{K}}^i, \hat{\boldsymbol{V}}^i \leftarrow \boldsymbol{K}^i_{\boldsymbol{C}_i}, \boldsymbol{V}^i_{\boldsymbol{C}_i}$
7 $\quad \boldsymbol{z}_j \leftarrow$ sample from LLM prediction
8 **return** $\{\boldsymbol{z}_j\}$

---

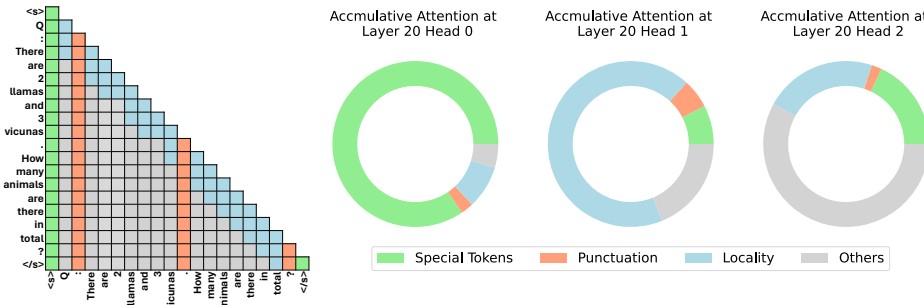

Figure 1: Different attention heads usually have different structures. Left: Four common attention structures (more details are elaborated in Section 3 and Section 4). Right: Attention map compositions of three attention heads that are in the same layer.

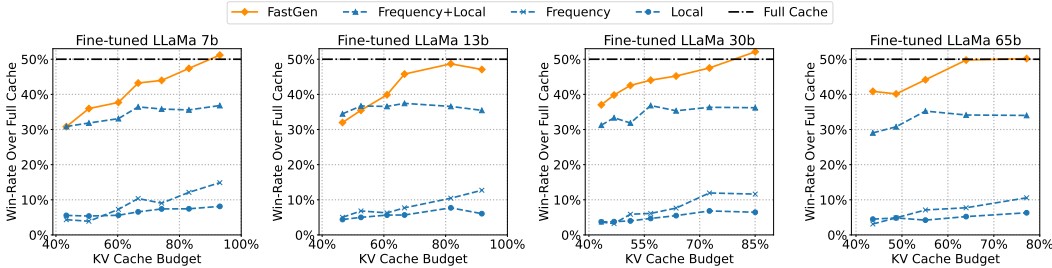

Figure 2: Performance of Adaptive KV Cache (FastGen) and Fixed KV Cache (Frequency, Local, and Frequency+Local; Zhang et al., 2023 and Liu et al., 2023a) on AlpacaEval.

In our study, FastGen recognizes five fundamental attention structures and applies them correspondingly. Specifically, some attention modules mostly attend to local contexts, for which we construct a KV cache that evicts long-range contexts; some primarily attend to specific tokens/punctuations, for which we create a KV cache that retains only special tokens/punctuations; some have attention maps that are column-wise sparse, for which we discard the least frequently attended tokens; and some broadly attend to all tokens, for which we employ the standard KV cache and store all tokens.

In this way, FastGen is able to compress the KV cache while retaining the original functionality of attention modules. Remarkably, FastGen does not require any fine-tuning and can be applied in a

plug-and-play manner. This is a big advantage of FastGen, because the training cost on extra-large models (Brown et al., 2020), can hardly be afforded by many research labs or practitioners.

We evaluate FastGen on Llama 1 (Touvron et al., 2023b) with a suite of major benchmarks covering generative tasks in math, code, knowledge, and common sense reasoning. FastGen effectively performs KV cache compression with negligible generation quality loss (i.e., recover over 95% of attention scores with 35% cache compressed). Notably, as to the 30b model in Figure 2, FastGen (50% cache compressed) surpasses all fixed KV compression methods (15% cache compressed).

## 2 RELATED WORK

**Token Dropping and KV Cache Compression.** Many efforts have been made to improve the model efficiency for LLMs. For recurrent neural networks, one method is to skip multiple tokens at a given time step (Campos et al., 2017; Seo et al., 2017; Hansen et al., 2019). Since Transformer models quickly attracted lots of attention, Goyal et al. (2020) proposes to eliminate redundant words in BERT (Devlin et al., 2019) based on their attention scores, while Dai et al. (2020) compresses the input sequence by adding pooling layers to the encoding modules of the transformer architecture. Recently, Huang et al. (2022) adds a token selection task to the original BERT model that learns to select performance-crucial tokens, and Kim et al. (2022) designs a learnable threshold to detect unimportant tokens to prune. Meanwhile, many efforts have been made to explore the possibility of compressing the hidden state of tokens rather than explicitly reducing the sequence length (Guan et al., 2022; Sun et al., 2022; Zhou et al., 2020).

Nevertheless, these methods can only be applied to non-autoregressive models and typically require an additional re-training phrase, making them less suitable for auto-regressive LLMs like ChatGPT and Llama. Recognizing this gap, researchers started examining the potential of pruning tokens within the KV cache of auto-regressive LLMs. Mu et al. (2023) learns to compress the prompts into a few special tokens to reduce memory pressure during caching. However, the token prediction requires model re-training and could be an expensive overhead during inference. Meanwhile, several concurrent methods propose to leverage accumulated attention score as the criteria to identify important tokens in the KV cache (e.g., Sheng et al., 2023; Zhang et al., 2023; Liu et al., 2023a). Instead of investigating a specific eviction policy, this study aims to synergistically coordinate diverse eviction policies to better align with model-specific attributes.

**Underlying Structure of Attention.** Inspired by the success of Transformer, extensive studies have been conducted to explore the underlying mechanism of different self-attention heads. Voita et al. (2019) analyzed the self-attention heads in BERT using LRF (Bach et al., 2015) and characterized them into interpretable roles, one of which is attending adjacent tokens all the time. Michel et al. (2019) demonstrated that heads in the same layer could have different impact on the performance while the importance of each head changes across tasks. Clark et al. (2019) and Kovaleva et al. (2019) identified such patterns as some heads primarily attend to separator tokens, adjacent tokens and a combination of these. While most previous studies mainly considered encoder models, FastGen is motivated by consistent patterns we have observed in decoder-only models. Like previous studies, FastGen also explores the structure of the attention mechanism to improve inference efficiency, but focusing on characterizing the KV cache of different attention heads.

## 3 ADAPTIVE KV CACHE COMPRESSION

In this section we first introduce the problem formulation, and then present attention profiling and adaptive KV cache compression.

### 3.1 GENERATIVE INFERENCE OF AUTOREGRESSIVE LLMS

A typical generative model inference involves two steps: prompt encoding and token generation.

**Prompt Encoding.** When an autoregressive transformer-based LLM generates the $i$-th token, the attention module needs to refer to all the preceding $i - 1$ tokens, i.e., the key and value vectors (KV vectors) of these tokens. To circumvent redundant KV vector computations when generating succeeding tokens, all KV vectors are stored in the *KV cache* once they are generated.

**Token Generation.** Once prompt encoding is finished, the LLM generates the output token by token. At each generation step, the LLM needs to encode the new token(s) generated in the previous step. After a new token is generated, its associated KV vectors are appended to the current KV cache. Thus, the size of KV cache increases linearly with the number of tokens being generated.

## 3.2 FASTGEN

As described in Section 2, many previous studies of compressing KV cache for improving inference efficiency do not leverage the intricate attention structure in LLMs. As to be detailed in Section 4, attention heads in LLMs often function distinctively, indicating the need for tailoring the compression strategy to each individual attention head.

With these insights, we introduce FastGen: a dual-phase algorithm for crafting an adaptive KV cache. During the prompt encoding phase, model profiling is conducted to discern the behavior of various attention heads, so that we can choose the most appropriate compression strategy for each head. Then, in the token generation phase, instead of indiscriminately appending new KV vectors for each newly generated token, we manage the KV cache based on the selected compression strategies.

## 3.3 MODEL PROFILING

Model profiling is conducted based on the result of prompt encoding. Specifically, for a compression policy $C$, we mark the corresponding KV cache compression as $K_C, V_C = f(K, V, C)$, where $K_C$ and $V_C$ are the compressed KV cache. Then, for attention map $A = \text{softmax}(QK^T)$, we pick the optimal policy that can recover $A$ with a recover ratio $T$ with the minimum memory cost:

$$C^* = \underset{C \in \mathcal{C}}{\arg\min} \ \text{CacheMemoryCost}(C) \quad \text{s.t.} \quad |A - \text{softmax}(QK_C^T)| \leq 1 - T, \quad (1)$$

where $\mathcal{C}$ is the set of all feasible compression policies, $\text{CacheMemoryCost}(C)$ is the target KV cache budget of the compression policy $C$, and $T$ a predefined hyper-parameter representing how much we want the policy to recover $A$. As to be discussed in Section 5, FastGen is able to recover +95% of the attention map with +40% compression ratio for a 65B model. The final prompt encoding algorithm that includes model profiling is presented in Algorithm 1.

Intrinsically, our method assumes that the structure of the attention map for a head is stable through the generation process. So, it is sufficient to use only the encoded prompt to select a proper compression policy. It is worth noting that existing literature has provided theoretical justifications for using solely encoded prompts to capture attention structures for the full contexts (Zhang et al., 2023; Liu et al., 2023a). In our study, we also empirically verified this, as to be elaborated in Section 4.

## 3.4 KV CACHE COMPRESSION POLICIES

In our experiments we observe that a large number of attention heads closely follow certain patterns, as to be detailed in Section 4. Thus, in addition to the conventional full KV cache policy, we also consider four fundamental KV cache compression policies. While we mainly use these four fundamental KV cache compression policies for evaluation in this study, it is easy for FastGen to use numerous other strategies. The four KV cache compression policies are:

- **Special Tokens.** We keep in KV cache only special tokens, such as the begin-of-the-sentence token , the instruction token [INST], and so on. This policy is referred to as $C_{\text{special}}$.

- **Punctuation.** We keep in the KV cache only punctuation tokens like ".", ":", "?". This policy is referred to as $C_{\text{punct.}}$.

- **Locality** This policy evicts long-range contexts. Once the relative distance between the context token and the current token exceeds a threshold, the KV cache of the context token will be evicted. The threshold is determined by a pre-defined ratio $r_l$ of the length budget of local context over the input sequence length. This policy is referred to as $C_{\text{local}}$.

- **Frequency (Heavy Hitter)** This policy has been used in multiple previous studies (e.g., Sheng et al., 2023; Zhang et al., 2023; Liu et al., 2023a). We monitor for each token its cumulative sum of attention score, then treat these scores as token *frequency* and only keep the most frequent

tokens in the KV cache. The length budget of frequent tokens over the current sequence length is controlled by a ratio $r_f$. This policy is referred to $C_{\text{frequent}}$.

**Hybrid Policies.** In practice, it is often necessary to use hybrid policies that combines the afore-mentioned compression policies. Since the total number of hybrid policies is hugh, in our study we use a greedy method to construct a small set of hybrid-policies as follows

$$\mathcal{C} = \{C_{\text{special}}, C_{\text{special+punct.}}, C_{\text{special+punct.+frequent}}, C_{\text{special+punct.+frequent+local}}, C_{\text{full}}\}, \quad (2)$$

where the sum of two compression strategies is to compute the union of their compressed KV cache, and $C_{\text{full}}$ refers to full KV cache without compression.

We use $C_{\text{special}}$ as a component in all hybrid policies for two reasons: 1) We observe that high attention scores are usually allocated towards $C_{\text{special}}$, as to be detailed in Section 4, indicating that $C_{\text{special}}$ are crucial for attention map recovery; 2) the compressed cache of $C_{\text{special}}$ is memory-efficient since there are usually less than 5 special tokens in a sentence. In other words, it brings little-to-no extra memory cost by always including $C_{\text{special}}$. Similarly, $C_{\text{punct.}}$ is often used as a component to form hybrid policies due to its memory-efficiency, i.e., the number of punctuations in a sentence is small. The final algorithm for token generation is presented in Algorithm 2.

## 4 DIVERSITY AND STABILITY OF ATTENTION STRUCTURES

In this section we present an empirical study to show the effectiveness of adaptive KV cache compression. First, we demonstrate that different attention heads typically possess distinct structures. Then, we show that these attention head structures remain relatively consistent across different attention heads at different positions. We do so by analyzing the attention scores of Llama 1 65B using random samples from GSM8k (Cobbe et al., 2021).

### 4.1 HEAD DISTINCTIVE ATTENTION STRUCTURE

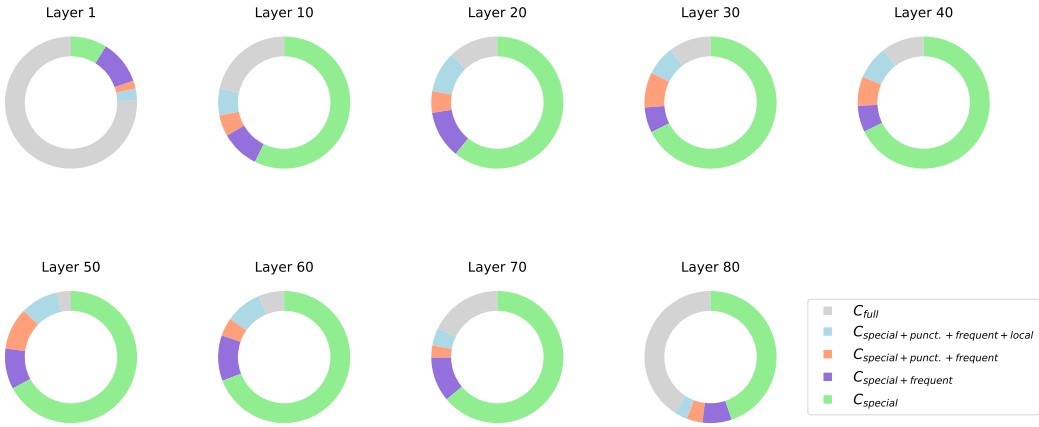

Figure 3: Attention profiling result distribution across different layers.

**Setting.** We perform model profiling with a recover threshold of $0.95$ and compute the distribution of profiling results for $\{1, 10, 20, 30, 40, 50, 60, 70, 80\}$ layers. The result is shown in Figure 3.

**Observation.** Figure 3 shows that attention heads in different layers have vastly different structures. Specifically, for the initial and final layers, they have more attention heads assigned to the full KV cache, indicating attention heads in these layers are likely to attend to all tokens. Meanwhile, for middle layers, the attention map focuses on special tokens, indicating that most attention heads of these layers primarily attend to special tokens (i.e., the accumulated attention score on special tokens is higher than $0.95$ for these attention heads). Figure 1 shows the structure of different attention heads in the same layer. We see that attention structures differ across different layers and heads.

These results indicate that it is suboptimal to apply the same KV cache to all layers without adaptation, and that it is beneficial to detect the structure of each attention head so as to select the optimal compression policy to construct the KV cache.

## 4.2 PROFILE TENDS TO BE CONSISTENT IN ONE SEQUENCE

The previous section demonstrates the great potential for constructing adaptive KV cache in accordance with the structure of different attention heads. Here, we show for each instance, it is sufficient to conduct one-shot model profiling, as outlined in Section 3.3. Specifically, for a given prompt, we show that the attention structure of each head remains consistent through the decoding process.

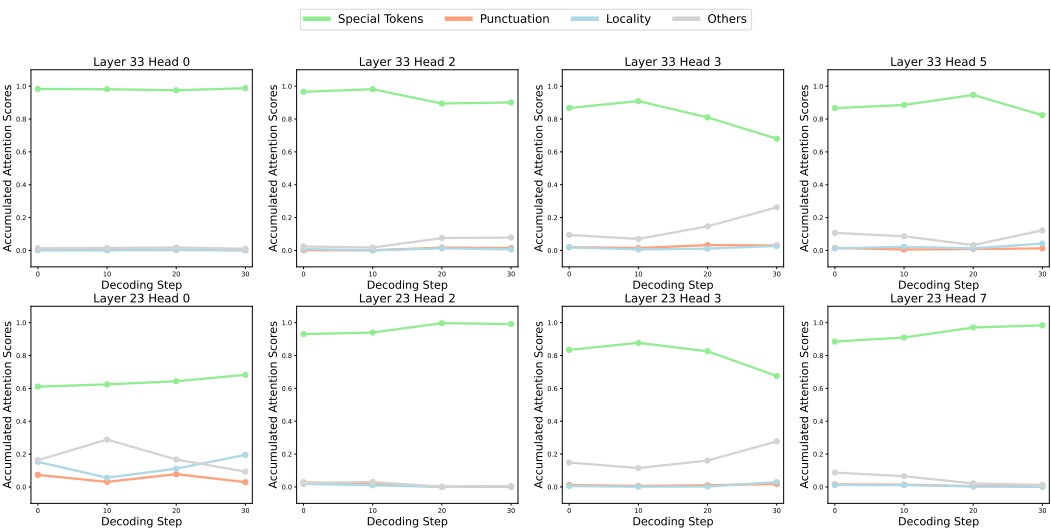

Figure 4: Accumulated attention score at 1st (prompt encoding), 10th, 20th, 30th decoding steps.

**Setting.** Following Figure 1, we compute the accumulated attention score for attention heads in different layers of Llama 1 65B at multiple decoding steps (i.e., 1st, 10th, 20th, 30th). We visualized the resulting accumulated score in Figure 4.

**Observation.** Despite some fluctuations of accumulated attention scores across time steps, the pattern of the attention maps remains relatively stable. For example, Layer 33 Head 0 and Layer 23 Head 2 almost only attend to the special token, while the locality and punctuation plays an important role in Layer 23 Head 0. As to Layer 23 Head 3, more than 10% of the attention score is allocated to the others portion, making it suitable for a uncompressed KV cache $C_{\text{full}}$.

In addition, we observe that a large portion of attention scores are on special tokens in all cases. This justifies the greed method we used to construct hybrid policies, as described in Section 3.4.

## 5 EXPERIMENT

We conduct comprehensive experiments to demonstrate the effectiveness of FastGen on memory footprint reduction and generation quality preserving. First, we report the trade-off between memory reduction and end-to-end generation quality in Section 5.1, and discuss the compression ratio of FastGen in Section 5.2. To demonstrate the superiority of FastGen on real-world systems, we demonstrate the end-to-end latency change in Section 5.3 and the profiling overhead in Section 5.4. Finally, we present ablation studies and discussions in Section A.1.

### 5.1 TRADE-OFF BETWEEN PERFORMANCE AND MEMORY REDUCTION

**Backbones.** We conduct experiments with both Llama 1 (Touvron et al., 2023a) and its fine-tuned variants, with model sizes ranging from 7B to 65B. For fined-tuned variants, we do not choose

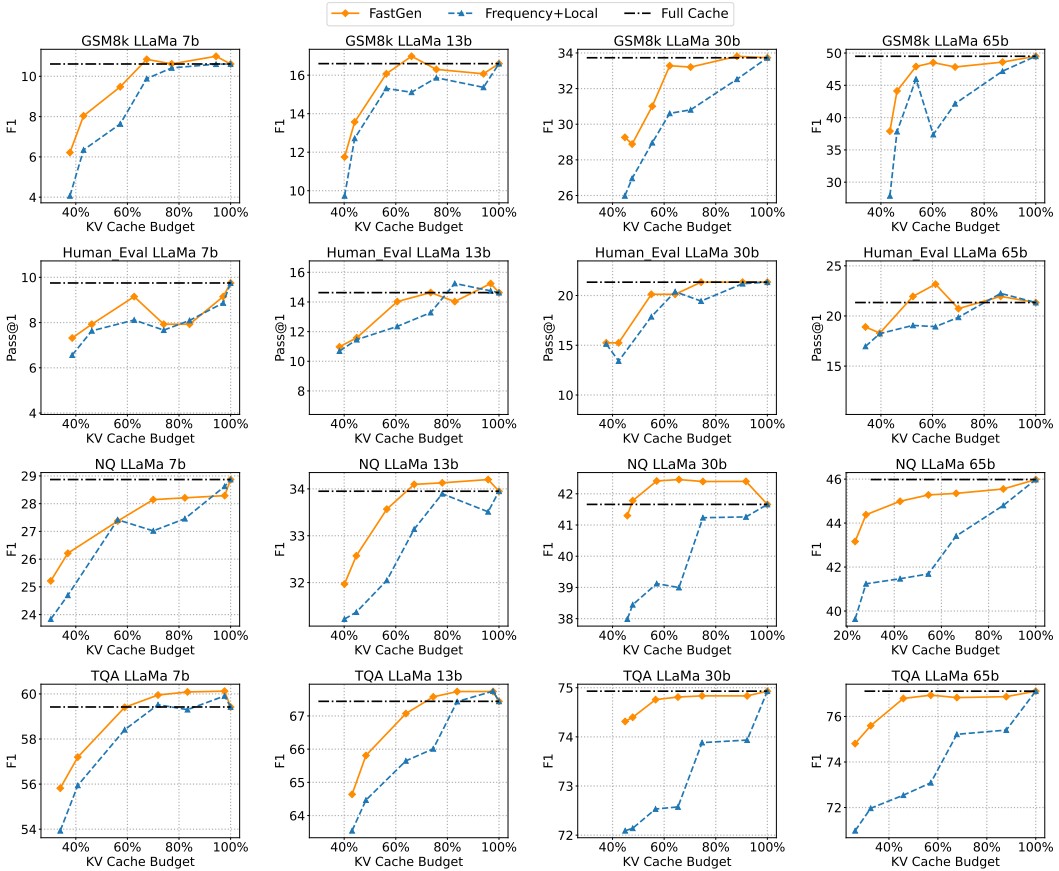

Figure 5: Performance of Adaptive KV Cache (FastGen) and Fixed KV Cache (Frequency+Local; Zhang et al., 2023 and Liu et al., 2023a) of Llama 1 on GSM8k, HumanEval, NQ, and TQA.

the open-sourced Llama 2-chat (Touvron et al., 2023b) model due to its grouped-query attention techniques. Instead, we use the original multi-head attention architecture in this study and leave the integration of grouped-query attention to future work. To prepare a comparable instruction-following model for analysis, we fine-tuned the Llama 1 model with open-sourced instruction-tuning datasets. Specifically, the fine-tuned variants are trained on LIMA[1] data (Zhou et al., 2023) and Open Assistant[2] (Köpf et al., 2023) data.

**Tasks.** We use standard generation tasks to evaluate Llama 1 and our fine-tuned Llama 1 models. For Llama 1, we choose 4 different tasks, including HumanEval (Chen et al., 2021), GSM8k (Cobbe et al., 2021), NQ (Kwiatkowski et al., 2019) and TQA (Kembhavi et al., 2017) to evaluate models' abilities on different domains (code, math, question answering and reading comprehension). Note that in the four tasks, each testing sample is in a generative format, where answers are extracted after model generation finishes. This is crucial for a fair comparison on model's generation quality. We evaluate the instruction finetuned LLaMa model on the instruction tuning benchmark AlpacaEval (Li et al., 2023), which consists of 805 question prompts from diverse domains.

**Experiment Setup.** The evaluation of the Llama 1 model follows the default setting and evaluation metrics on each benchmark. We calculate F1 scores for GSM8k, NQ and TQA, and use the code execution Pass@1 rate for HumanEval. While evaluating an instruction-tuning model remains challenging, we follow previous work (Zhou et al., 2023; Touvron et al., 2023b) to use GPT4 as an evaluator for pair-wise comparison between two different model generations. For each prompt,

---

[1]https://huggingface.co/datasets/GAIR/lima.

[2]https://huggingface.co/datasets/OpenAssistant/oasst1.

we input the FastGen generation and the generation from the same model with Full KV Cache as a pair, and ask GPT4 to judge which one is better. We then calculate the win rate of FastGen over Full Cache. Hypothetically, the win rate of a lossless method should be around 50%. Aside from full-cache models, we also include non-adaptive KV cache methods for comparison. Specifically, we apply $C_{local}$, $C_{frequent}$, and $C_{local+frequent}$ to all attention head without any adaptation, as baselines. It is worth mentioning that $C_{local+frequent}$ is a very strong baseline as it is identical to the H$_2$O method (Zhang et al., 2023) and the Scissorhands method (Liu et al., 2023a). We set $r_l = 0.3$, $r_f = 0.3$ in FastGen, and only change the recovery ratio $T$ to control the pruned KV cache ratio. For generation, we use nucleus sampling (Holtzman et al., 2019) with temperature T = 0.6, p = 0.9. Experiments are conducted on 8 NVIDIA A100 80GB GPUs.

**Main Results.** In Figure 2 and Figure 5, we present the model quality as a function of KV cache budget increasing from $30\%$ to $100\%$. For 30B models, FastGen (50% cache compressed) surpasses all non-adaptive KV compression methods (15% cache compressed) . Also, we can see FastGen achieves more KV cache reduction ratio as the model size increases, while preserving the same model quality. For example, achieving a $45\%$ win rate, FastGen can get as much as 44.9% pruned ratio on Llama 1-65B, compared to 16.9% pruned ratio on Llama 1-7B. In all settings, FastGen shows consistent and significant improvement over non-adaptive compression methods. The results validate the effectiveness of adaptive KV cache compression using FastGen, despite its simplicity.

| Model Size | KV Cache | | | $T$ | Win rate |
|---|---|---|---|---|---|
| | Full | FastGen | Pruned ratio | | |
| 7B | 4.3Gb | 1.9Gb | 56.6% | 91% | 30.8% |
| | | 2.6Gb | 39.8% | 95% | 37.7% |
| | | 3.6Gb | 16.9% | 98% | 47.4% |
| 13B | 6.7Gb | 3.1Gb | 53.4% | 91% | 32.0% |
| | | 4.1Gb | 39.0% | 95% | 39.9% |
| | | 5.5Gb | 18.3% | 98% | 48.7% |
| 30B | 13.1Gb | 5.7Gb | 56.7% | 93% | 37.0% |
| | | 6.7Gb | 48.8% | 95% | 42.5% |
| | | 9.5Gb | 27.4% | 98% | 47.5% |
| 65B | 21.5Gb | 9.4Gb | 56.3% | 93% | 40.9% |
| | | 11.8Gb | 44.9% | 95% | 44.2% |
| | | 13.8Gb | 36.0% | 98% | 49.8% |

Table 1: Memory footprint reduction by FastGen. We compared the memory consumption between models with full KV cache, and models compressed by FastGen on fine-tuned Llama 1.

## 5.2 MEMORY FOOTPRINT REDUCTION ANALYSIS

We report the KV cache memory footprint reduction in Table 1. For all the evaluated 7B-65B models, we evaluate the memory consumption with a fixed batch size of 16, sequence length of 512, and model weights in fp16 format. We observe that FastGen substantially reduces the KV cache memory footprint across all model sizes, with more significant reductions for larger models. Taking a win rate over 45% as little-to-no quality regression, FastGen can achieve ∼40% memory reduction in Llama 1-65B, ∼30% in Llama 1-30B, ∼20% in Llama 1-13B and Llama 1-7B.

## 5.3 END-TO-END LATENCY IMPROVEMENT

Table 2: End-to-end latency comparison on Llama 1-7B.

| Batch size | 1 | | | | 2 | | | 8 | | 16 |
|---|---|---|---|---|---|---|---|---|---|---|
| [prompt len, gen len] | [32,512] | [32,2048] | [32,8192] | [32,16384] | [512,32] | [512,512] | [4096,4096] | [512,512] | [4096,4096] | [512,512] |
| HF | 13.35 | 57.37 | 299 | 799.14 | 1.12 | 19.16 | 167.64 | 23.44 | OOM | OOM |
| DS | 11.58 | 47.12 | 201.23 | 435.74 | 0.79 | 10.45 | 91.04 | 12.93 | 127.94 | OOM |
| FastGen | 11.21 | 44.6 | 179.43 | 359.83 | 0.73 | 9.71 | 76.93 | 10.57 | 82.16 | OOM |
| Speed-up(%) over HF | 16.03% | 22.30% | 40.00% | 55.00% | 34.80% | 49.30% | 54.10% | 54.90% | - | OOM |
| Speed-up(%) over DS | 3.20% | 5.35% | 10.83% | 17.42% | 7.59% | 7.08% | 15.50% | 18.25% | 35.78% | OOM |

To analyze the end-to-end speedup of FastGen, we present the end-to-end latency improvement over full-cache setting and a strong model acceleration baseline in Table 2. In the experiment, we record the total duration in seconds, measured from the start of prompt encoding, until the end of generation as the end-to-end latency. For the full-cache baseline, we adopt the widely used Hugging Face Accelerate (HF) (Gugger et al., 2022), denoted as HF in Table 2. For FastGen, we implemented a customized kernel to handle the KV cache pruning operation. Specifically, we adapt the kernel from Deepspeed (DS) (Aminabadi et al., 2022) by adding the KV cache sparsity operation. We include the Deepspeed performance for fair comparison, denoted as DS in Table 2. All methods are tested on the same Nvidia V100 GPUs.

As shown in Table 2, we can observe that FastGen achieves significant end-to-end speed-up across all the generation settings. For the least significant case, FastGen can have a decent $16.04\%$ latency improvement over the HF baseline on a short generation length of 512. In the best cases, we can achieve up to $55.0\%$ latency reduction over HF with FastGen at a generation length of 16k. We can also observe that the relative speedup is greater with longer generation length. For example, given batch size = 1, FastGen's relative speed-up rises from $16.04\%$ to $55.0\%$, as the generation length grows from 512 to 16k. When comparing FastGen to DeepSpeed, we can still observe significant speed-up that gets bigger with batch size and generation length. Considering DeepSpeed is a full-stack optimized inference system, where not only attention computation is optimized, there is still much room to further improve FastGen by polishing the sparsity kernel. We leave this unique research and engineering challenge to future works.

## 5.4 PROFILING COST

Table 3: Profiling time of Llama 1-65B. The Overall Generation Duration is measured from the start of decoding to the end of the generation length. The Profiling Duration is measured from the start of the decoding until Fastgen finishes the policy search.

| Generation Length | Overall Generation Duration (s) | Profiling Duration (s) | Decoding Time Per Token (s) | Profiling/Overall (%) |
|---|---|---|---|---|
| 128 | 30.98 | 0.11 | 0.10 | 0.35% |
| 256 | 50.1 | 0.11 | 0.10 | 0.21% |
| 512 | 94.98 | 0.11 | 0.10 | 0.12% |
| 1024 | 157.43 | 0.11 | 0.10 | 0.07% |

To better understand the overhead of the profiling step, we compare the profiling time with the total generation time across different generation lengths. We present the result in Table 3.

We can observe that the profiling time only accounts for a very small percentage of the total generation duration, up to $0.35\%$ in our tested cases. Also, the overhead decreases as the generation length increases, dropping to $0.07\%$ when the generation length comes to 1024.

In terms of extra memory usage, it's mainly introduced by one of the compression strategies, $C_{\text{frequent}}$, which needs to store an extra cumulative sum of attention scores for each attention head. To provide a detailed analysis, for each layer, the dimension of the KV cache is (`batch_size`, `num_of_head`, `sequence_len`, `hidden_dimension`), while the dimension of extra memory for the cumulative attention scores is (`batch_size`, `num_of_head`, `sequence_len`). Considering `hidden_dimension` = 128 for all model sizes, the memory overhead is $1/128$=0.78% compared to storing KV cache only, which is a negligible cost.

## 6 CONCLUSION

We have presented FastGen, a novel method that significantly improves the inference efficiency of LLMs, with no visible quality loss, using lightweight model profiling and adaptive key-value caching. Areas for future explorations include combining FastGen with other model compression techniques, such as quantization and distillation, and other efficient attention architectures, such as grouped-query attention.

ACKNOWLEDGMENTS

Research was supported in part by US DARPA KAIROS Program No. FA8750-19-2-1004 and IN-CAS Program No. HR001121C0165, National Science Foundation IIS-19-56151, and the Molecule Maker Lab Institute: An AI Research Institutes program supported by NSF under Award No. 2019897, and the Institute for Geospatial Understanding through an Integrative Discovery Environment (I-GUIDE) by NSF under Award No. 2118329. Any opinions, findings, and conclusions or recommendations expressed herein are those of the authors and do not necessarily represent the views, either expressed or implied, of DARPA or the U.S. Government.

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

# A APPENDIX

## A.1 ABLATIONS

For all the ablations, we use a fixed targeted recovery ratio T = 0.98.

**How one policy affect all the other policies?** We study the complementary effects of each policy on the combination of all other policies in our framework. We examine changes in pruned KV cache and win rate while fixing the targeted recovery ratio $T$. We take the full policy set as our control set $\mathcal{C}$. For each ablation, we remove one of the policies from all policy combination in $\mathcal{C}$. We summarized the results in Table 4, which suggests the $C_{\text{frequent}}$, and the $C_{\text{special}}$ are the most important policies. Removing them will incur a 3.67% and a 2.11% win rate drop respectively. We can also observe from the pruned cache ratio that $C_{\text{frequent}}$ and $C_{\text{local}}$ reduce more KV caches than the others. However, their standalone non-adaptive deployment yields suboptimal performance, as depicted in Figure 2, further verifying the importance of adapting different compression policies.

| Feasible Policy Set | Pruned KV Ratio | Win Rate |
|---|---|---|
| $\mathcal{C}$ | 36.04% | 49.75% |
| $\{C_{\text{punct.}}, C_{\text{punct.+frequent}}, C_{\text{punct.+frequent+local}}, C_{\text{full}}\}$ | 31.16% | 47.64% |
| $\{C_{\text{special}}, C_{\text{special+frequent}}, C_{\text{special+frequent+local}}, C_{\text{full}}\}$ | 34.23% | 49.56% |
| $\{C_{\text{special}}, C_{\text{special+punct.}}, C_{\text{special+punct.+frequent}}, C_{\text{full}}\}$ | 30.18% | 49.06% |
| $\{C_{\text{special}}, C_{\text{special+punct.}}, C_{\text{special+punct.+local}}, C_{\text{full}}\}$ | 21.26% | 46.08% |

Table 4: Complementary effects of each policy. We display the win rate of each method over full cache setting. We evaluate the fine-tuned Llama 1-65B on AlpacaEval with the same parameters.

| Cache Order | Pruned KV Ratio | Win Rate |
|---|---|---|
| $C_{\text{special}} \rightarrow C_{\text{punct.}} \rightarrow C_{\text{frequent}} \rightarrow C_{\text{local}}$ | 36.04% | 49.75% |
| $C_{\text{special}} \rightarrow C_{\text{frequent}} \rightarrow C_{\text{local}} \rightarrow C_{\text{punct.}}$ | 36.40% | 47.64% |

Table 5: Policy order ablation on fine-tuned Llama 1-65B with AlpacaEval.

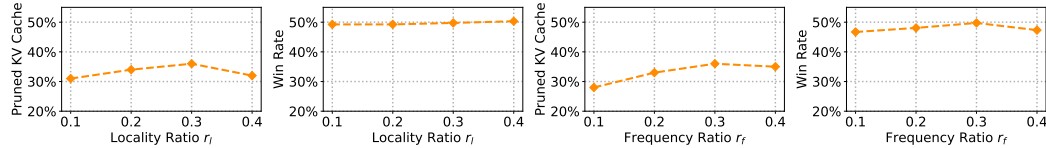

Figure 6: Hyper-parameter ablation on fine-tuned Llama 1-65B with AlpacaEval.

**Which policy should we add first (and last)?** As in Section 3.4, we use a greed method to construct adaptive KV cache. Here, we examine how the order of introducing each policy affects the performance. Similar to the previous study, we fix the targeted recovery ratio to 0.98, and keep allocating cache budget until the constructed cache hit the recovery ratio. For simplicity, we make every examined order opt-in the $C_{\text{special}}$ first, as it's typically the most important tokens and of super-low memory cost, as suggested in Figure 1. We summarize the results in Table 5. Our current order (as in Equation 2) achieves the highest win-rates and the highest pruned ratios. Meanwhile, using alternative orders leads to a different trade-off between KV cache compression and generation quality. For example, using $C_{\text{frequent}} \rightarrow C_{\text{local}} \rightarrow C_{\text{punct.}}$ leads to an improved KV cache compression ratio at the cost of generation quality.

## A.2 SENSITIVITY STUDY.

We analyze the sensitivity of selecting different hyper-parameters for FastGen, as illustrated in Figure 6. We observe that altering these hyper-parameters does not have a visible impact on the generation quality, as the model maintains a winrate over 45% in all situations. Meanwhile, it leads to a relative large change on the compression ratio. For example, changing the ratio for the frequency policy from 0.3 to 0.1 leads to more KV cache. In our experiments, we set the ratio to 0.3 for both $r_l$ and $r_f$.

