# OpenReview forum: "Model Tells You What to Discard: Adaptive KV Cache Compression for LLMs"
_ICLR.cc/2024/Conference — ICLR 2024 oral_

### Official Review · Reviewer_XtqU · 2023-10-26

**Soundness:** 3 good
**Presentation:** 3 good
**Contribution:** 3 good
**Rating:** 8
**Confidence:** 4

**Summary:**

Key-value cache takes the majority of GPU memory in LLM serving, and its extent is continuously growing along with the model size and context length. Therefore, if we can reduce the key-value cache memory while maintaining the generation quality, we can accelerate the LLM inference. The authors propose FastGen, a framework for efficient generative inference by applying on-the-fly key-value compression. They analyzed the structural patterns of each attention head of layers and then categorized four policies. By adopting the optimal policy based on profiling, FastGen achieves a comparable generation quality to the full-cache (non-compression) inference.

**Strengths:**

- The authors provide abundant experiments with varying model sizes and tasks
- The authors provide informative ablation studies
- Their work will motivate various related work, for example,
    - efficient kernel which aware of compression
    - as the model size grows, more redundant key-values exist where we have more room to optimize

**Weaknesses:**

- Since the inference time matters in practical serving, it would be helpful to understand more if the authors can provide corresponding results
    - For example, how long does inference take compared to the full-cache strategy? I think it might become slower because the existing attention kernels may not efficiently deal with the sparsity
    - How long does profiling take? Is it feasible for practical inference scenarios?
- It seems that the StreamingLLM paper [1] is similar to this work. It sets sink tokens and performs local attention, where the sink tokens may correspond to $C_{special}$ (and maybe $C_{punct}$. Since the StreamingLLM paper has also recently been uploaded, it is unlikely to compare this paper with it. But it would be better if the differences in this paper were clarified.

[1] Xiao, Guangxuan, et al. "Efficient Streaming Language Models with Attention Sinks." arXiv preprint arXiv:2309.17453 (2023).

**Questions:**

- What are the additional challenges for the models that use the grouped query attention technique?
- In Figure 4, attention scores of special tokens always take more than half. Are there attention heads whose special token score is lower than half?
- In Figure 5, compressing sometimes wins the full-cache strategy. How can we interpret such results?

---

> ### Author Response · Authors · 2023-11-23
>
> **W1.1: How long does inference take compared to the full-cache strategy? I think it might become slower because the existing attention kernels may not efficiently deal with the sparsity.**
>
> R1.1: To address reviewers’ concerns on the end-to-end speedup of FastGen, we implement a sparsity kernel for KV-cache pruning and present the end-to-end latency improvement in Table 1. Please refer to general response #1 for detailed settings and analysis.
>
> As shown in Table 1, we can observe that FastGen achieves significant end-to-end speed-up across all the generation settings. For the least significant case, Fastgen can have a decent 16.04% latency improvement over the full-cache baseline on a short generation length of 512. In the best cases, we can achieve up to 55.0% latency reduction with Fastgen at a generation length of 16k.
> We can also observe a clear and consistent tendency of larger relative speed-up as generation length becomes longer. For example, given batch_size = 1, FastGen’s relative speed-up rises from 16.04% to 55.0%, as the generation length grows from 512 to 16384. The phenomenon can also be observed in other batch settings.
>
> This analysis confirms that FastGen can achieve major speed-up in real development, especially in long-generation settings. Meanwhile, the efficiency of the customized kernels can be further improved. We leave this unique research and engineering challenge to future works.
>
> **W1.2: How long does profiling take? Is it feasible for practical inference scenarios?**
>
> R1.2: To better understand the overhead of the profiling step, we compare the profiling time with the total generation time across different generation lengths. We present the result in Table 2. Please refer to general response #2 for detailed settings and analysis.
>
> Table 2 shows the profiling time of the LLaMA65b in different generation length settings. We can observe that the profiling time only accounts for a very small percentage of the total generation duration, up to 0.35% in our tested cases. Also, the overhead decreases as the generation length increases, dropping to 0.07% when the generation length comes to 1024.
>
> In terms of extra memory usage, it’s mainly introduced by one of the compression strategies, C_frequent, which needs to store an extra cumulative sum of attention scores for each attention head.  To provide a detailed analysis, for each layer, the dimension of the KV cache is (batch_size, num_of_head, sequence_len, hidden_dimension), while the dimension of extra memory for the cumulative attention scores is (batch_size, num_of_head, sequence_len). Considering hidden_dimension=128 for all model sizes, the memory overhead is 1/128=0.78% compared to storing KV cache only, which is a negligible cost.
>
> In conclusion, the overhead introduced by the profiling step is nearly negligible in both time and memory, which confirms FastGen’s potential for real-world deployment.
>
> **W2: It seems that the StreamingLLM paper [1] is similar to this work.**
>
> A2: Thanks for mentioning this concurrent work! We were not aware of it at the time of submission given it was posted after the ICLR deadline. After reading the paper, we think it is a great parallel work on long-context LLM, and we are happy to provide a comparison between it and FastGen! Generally, the two differ in two aspects:
> 1) Goal and Setting: StreamingLLM aims to extend the context length of LLM, while FastGen aims to improve the efficiency of general LLM inference (normal+long contexts). As a result, FastGen focuses specifically on generation tasks, which is different from the general task settings in StreamingLLM.
> 2) Method: StreamingLLM uses a fixed attention pruning strategy for all attention heads, while FastGen focuses on adaptively choosing different compression strategies according to the attention structure of each head.

---

> > ### Author Response · Authors · 2023-11-23
> > **(Continued) Official Comment by Authors**
> >
> > **Q1: What are the additional challenges for the models that use the grouped query attention technique?**
> >
> > A1: Thanks for mentioning grouped query attention (GQA). We are working on accommodating FastGen to GQA. In GQA, heads within each group share the same KV vectors. Instead of head-wise pruning, we could modify FastGen to perform group-wise pruning. Specifically, we could individually evaluate each query by calculating the recovery ratio of its attention map (Q*K), and then average ratios from all heads within the same group, using the averaged ratio as the criteria to find the optimal strategy for each group. As analyzed, adapting FastGen to GQA requires minimal methodology change and imposes little additional cost. We are working on obtaining concrete experimental results to show the effectiveness of GQA.
> >
> > **Q2: In Figure 4, attention scores of special tokens always take more than half. Are there attention heads whose special token score is lower than half?**
> >
> > A2: This is a very good observation. There are attention heads with special token scores less than half. In Figure 4, we only show two specific layers, i.e., 23 and 33. However, in Figure 3, we could find layers without special tokens as the dominant type, e.g., layer 1 and layer 80.
> >
> > **Q3: In Figure 5, compressing sometimes wins the full-cache strategy. How can we interpret such results?**
> >
> > A3: Thank you for mentioning this, we think the performance instability could be attributed to two aspects:
> > 1): Figure 5 presents win-rate changes, which are calculated from GPT-4’s pair-wise voting and thus could fluctuate due to the uncertainty in GPT-4 generation sampling.
> > 2): Small perturbations in KV cache may add some uncertainty to model performance, which could sometimes improve robustness and lead to slightly higher results.

---

### Official Review · Reviewer_rrTE · 2023-10-29

**Soundness:** 3 good
**Presentation:** 2 fair
**Contribution:** 3 good
**Rating:** 8
**Confidence:** 3

**Summary:**

This paper propose an adaptive KV cache compression technique to reduce the memory footprint of generative inferences of LLMs. The authors fist perform targeted profiling to indentify the intrinsic structure of attention modules, and then build an adaptive KV cache by evicting long-range contexts on attention heads emphasizing local contexts, removing non-special tokens on attention heads centered on special tokens, and using only the standard KV cache. The experimental results show the adaptive KV cache achieves large reduction on GPU memory consumption with trivial geneation quality loss.

**Strengths:**

1. The paper works on an important topic, i.e., reducing the memory footprint of GPU during generative inferneces of LLMs.
2. The paper flows well.

**Weaknesses:**

1. The model profiling part is not clear. Did the authors do a profiling for each model on all datasets, or each model on a single dataset.
2. The model profiling results have a huge impact on the final KV cache compression results. Although the authors show empirical data supporting the structure of the attention map is stable at different positions for all attention heads, the authors still need to discuss what if the structure of the attention map is not stable.

**Questions:**

Please comment the two points in the weakness section.

---

> ### Author Response · Authors · 2023-11-23
> **Response to Reviewer rrTE**
>
> **W1: The model profiling part is not clear. Did the authors do a profiling for each model on all datasets, or each model on a single dataset.**
>
> A1: The profiling is run on every data instance (each sentence). For the same model, the profiling would be different if the input is different. So, we do the profiling on the fly during deployment. Such fine-grained flexible adaptation allows FastGen to reduce more memory footprint while preserving the model quality. We also provide more analysis on the overhead of profiling in Table 2 of **General Response**, which shows the cost is nearly negligible.
>
> **W2: the authors still need to discuss what if the structure of the attention map is not stable.**
>
> A2 We are not quite clear what the reviewers mean by "the structure of the attention map is not stable". We hypothesize that the reviewer was referring to the results in Figure 4, which shows the accumulated attention scores remain consistent/stable during the entire generation phase. The reviewer might be wondering if this observation can be generalized to all situations (e.g., different datasets and models). If that's the case, we think the reviewer raises a valid concern about FastGen, as some strategies such as static eviction policy (e.g., punctuation tokens) may no longer provide an accurate prediction, and more adaptive policies are needed.  To remedy this, we propose to repeat the diagnosis step multiple times across the generation process of one sentence. To be more specific, we choose a new pruning policy once the number of decoded tokens reaches $k$, where $k \in$[1,sentence_len] is a predefined hyperparameter. For each sentence, the diagnosis is repeated for sentence_len//$k$ times. From the analysis in the second part of General Response, we could infer that the profiling overhead is relatively small even if it is repeated several times. On the other hand, it would be interesting to study the theoretic explanation of our observation, which we would like to explore in the future.

---

> ### Comment · Area_Chair_agrp · 2023-12-04
> **[Important] Response Required to Authors' Rebuttal**
>
> Dear Reviewer rrTE,
>
> As we progress through the review process for ICLR 2024, I would like to remind you of the importance of the rebuttal phase. The authors have submitted their rebuttals, and it is now imperative for you to engage in this critical aspect of the review process.
>
> Please ensure that you read the authors' responses carefully and provide a thoughtful and constructive follow-up. Your feedback is not only essential for the decision-making process but also invaluable for the authors.
>
> Thank you,
>
> ICLR 2024 Area Chair

---

### Official Review · Reviewer_9wxD · 2023-10-30

**Soundness:** 3 good
**Presentation:** 3 good
**Contribution:** 3 good
**Rating:** 8
**Confidence:** 3

**Summary:**

This paper proposes FastGen, an adaptive key-value (KV) cache compression method to reduce the memory footprint and accelerate inference for large language models (LLMs). The key ideas are: 1) Profiling attention modules to discern their intrinsic structures, such as primarily attending to local contexts or special tokens. 2) Constructing the KV cache adaptively based on the recognized structure to compress less useful contexts. 3) The lightweight attention profiling guides the KV cache compression without expensive fine-tuning.

The experiments are conducted on LLaMa models with sizes from 7B to 65B parameters on diverse generative tasks. Results show FastGen effectively compresses the KV cache to 40-50% smaller with negligible quality loss. It also outperforms non-adaptive baselines.

**Strengths:**

Strengths:
-----------------
+ Adaptively compressing KV cache better aligns with model-specific attributes without retraining.
+ Comprehensive experiments verify FastGen works for diverse models and tasks. Up to 50% compression on 65B LLaMa with little quality loss is remarkable.
+ Ablation studies provide good insight into the design choices. The profiling method and compression policies are well motivated.

**Weaknesses:**

Weaknesses:
-----------------
- The compression policies are combined in a naive way. More advanced adaptive selection could be explored (see detailed in C1).
- No experiment on encoder-decoder models. The efficacy on them is unclear (see detailed in C2).
- More analysis on the overhead of profiling could be provided (see detailed in C3).

**Questions:**

Comments:
-----------------
C1:	The compression policies are combined in a simple naive way in FastGen, by just taking the union of multiple policies such as Cspecial + Cpunct + Cfrequent. This straightforward combination approach has several potential issues. First, is it possible that the union combination may introduce redundancy, as different policies could select overlapping important content, leading to suboptimal compression ratios? More intelligent strategies should consider the complementarity between modules to avoid duplicating the key contexts. Second, is it possible that existing policies may not be fully compatible? Some combinations could introduce conflicts and hurt generation quality. More systematic analysis should examine the compatibility between policies.

C2:	The experiments in the paper are all conducted on the decoder-only LLaMa models, without validation on encoder-decoder models like BART and T5. These models are also widely used for generative tasks, so the efficacy of FastGen on them remains unclear. This is worth further investigation.

C3:	The paper lacks sufficient analysis on the overhead and time cost of conducting attention profiling, which is important to judge the efficiency of FastGen in real deployment. Specifically, the time complexity of attention profiling needs analysis, and concrete profiler time under different model sizes should be provided or disscussed. Moreover, analyzing the extra memory or GPU memory required for the profiler and assessing its impact on deployment is necessary. In summary, quantitatively analyzing the resource overhead for profiling and demonstrating effective solutions to reduce it could strengthen the practicality of FastGen in real-world usage. Further experiments on optimized profiling and its cost-benefit trade-off with compression performance could provide more comprehensive insights into the efficacy of the approach.

---

> ### Author Response · Authors · 2023-11-23
>
> **W1: The compression policies are combined in a simple naive way in FastGen. This straightforward combination approach has several potential issues. First, is it possible that the union combination may introduce redundancy, as different policies could select overlapping important content? Second, is it possible that existing policies may not be fully compatible? Some combinations could introduce conflicts and hurt generation quality.**
>
> A1: We agree that different compression strategies have overlapping tokens. For example, C_frequent is sometimes overlapped with C_special and C_local, as special tokens and local contexts usually accumulate higher attention scores. However, FastGen is designed to progressively evict tokens. When there is an overlapped policy, we only consider the complementary tokens it contains. It is guaranteed that unioning the policy monotonously will only bring in newly evicted tokens.
> We further confirm that different policies are compatible in the ablation study (section 5.3). We study (1) how removing one strategy and (2) how changing relative policy order affects the overall performance.
>
> We can draw an empirical conclusion that little-to-no conflicts exist between different policies. In fact, we observe complementary effects between existing strategies. That is to say, the performance of standalone strategies can usually be boosted significantly by introducing other strategies when necessary.
>
> Some combinations can introduce inferior/superior performance than others. In our experiment, over all datasets, we find that following the order of  C_special, C_punct, C_frequen, C_local consistently gets the best performance. We leave investigating the intrinsic mechanism to future work.
>
> **W2: The experiments in the paper are all conducted on the decoder-only LLaMa models, without validation on encoder-decoder models like BART and T5.**
>
> A2: Thanks for the suggestion, FastGen could be easily adapted to encoder-decoder models by pruning the KV cache in their decoder. We will elaborate on this in the introduction and add several related works in the future version. In encoder-decoder models, KV cache is still used in the decoder to save computation. During generation, it is a standard practice to fix the encoder inputs as existing prompts or instructions, e.g., as in BART [1] and FlanT5 [2]. In such scenarios, the decoder part still works autoregressively to output newly generated tokens, and it is flexible to directly apply FastGen to their KV cache. In this paper, we focus on decoder-only models as most of the prevailing LLMs are decoder-only models, e.g., LLaMa, OPT, and GPT. We agree that additional discussions and experiments should be added, and we will revise accordingly.
>
> **W3: The paper lacks sufficient analysis on the overhead and time cost of conducting attention profiling, which is important to judge the efficiency of FastGen in real deployment.**
>
> A3: Thanks for the valuable advice. We provide extra experimental results on book-keeping overhead in Table 2. Please refer to the general response #2 for a detailed time and memory analysis of profiling cost.
>
> In short, Table 2 shows the profiling time of the LLaMA65b in different generation length settings. We can observe that the profiling time only accounts for a very small percentage of the total generation duration, up to 0.35% in our tested cases. Also, the overhead decreases as the generation length increases, dropping to 0.07% when the generation length comes to 1024.
>
> In terms of extra memory usage, it’s mainly introduced by one of the compression strategies, C_frequent, which needs to store an extra cumulative sum of attention scores for each attention head.  To provide a detailed analysis, for each layer, the dimension of the KV cache is (batch_size, num_of_head, sequence_len, hidden_dimension), while the dimension of extra memory for the cumulative attention scores is (batch_size, num_of_head, sequence_len). Considering hidden_dimension=128 for all model sizes, the memory overhead is 1/128=0.78% compared to storing KV cache only, which is a negligible cost.
>
> In conclusion, the overhead introduced by the profiling step is nearly negligible in both time and memory, which confirms Fastgen’s potential for real-world deployment. We additionally provide end-to-end system latency improvement in Table 1. It shows that FastGen can achieve major speed-up in various generation settings. Please refer to General Response #1 for more analysis.
>
>
>
> **Reference:**
>
> [1] Lewis, Mike, et al. "BART: Denoising Sequence-to-Sequence Pre-training for Natural Language Generation, Translation, and Comprehension." Proceedings of the 58th Annual Meeting of the Association for Computational Linguistics. 2020.
>
> [2] Chung, HyungWon et al. "Scaling Instruction-Finetuned Language Models." arXiv preprint arXiv:2210.11416(2022).

---

> ### Comment · Area_Chair_agrp · 2023-12-04
> **[Important] Response Required to Authors' Rebuttal**
>
> Dear Reviewer 9wxD,
>
> As we progress through the review process for ICLR 2024, I would like to remind you of the importance of the rebuttal phase. The authors have submitted their rebuttals, and it is now imperative for you to engage in this critical aspect of the review process.
>
> Please ensure that you read the authors' responses carefully and provide a thoughtful and constructive follow-up. Your feedback is not only essential for the decision-making process but also invaluable for the authors.
>
> Thank you,
>
> ICLR 2024 Area Chair

---

### Official Review · Reviewer_BiHE · 2023-10-30

**Soundness:** 4 excellent
**Presentation:** 3 good
**Contribution:** 4 excellent
**Rating:** 8
**Confidence:** 4

**Summary:**

The paper addresses the memory footprint reduction of LLMs during inference, in which the recent problem is the KV cache eviction/compression policies. The paper proposes an adaptive KV cache compression technique that operates in two stages, i) diagnose through profiling based on the attention heads and ii) applying an eviction strategy per each layer.

**Strengths:**

- Having an adaptive KV cache for each of the attention module type is a really interesting and exciting idea.
- No fine-tuning costs of the proposed method is commendable.
- The paper clearly positions within the body of existing literature, by distinguishing the proposed method as an adaptive and a diverse set of eviction strategies.
- The paper is clearly written, the presentation is great, easy to follow along and digest the concepts.

**Weaknesses:**

- Although, the idea of adaptive KV cache compression sounds interesting, what is the overhead of book-keeping to support this adaptive and diverse ability based on the type of the attention? This is not discussed anywhere in the paper?
  - That is, each layer id will be mapped to a eviction policy and is deployed with the model at hand.
  - Next, what is the added computational complexity both asymptotically as well experimentally.
- Table 3 shows an ablation on the policy order, why is this needed? Is the policy fixed per layer and the order will be dictated by the layer that needs a certain policy determined by the diagnosis step. Is it not true, clarify on this please.
- Another interesting exploration/ablation to see is to experiment with long context tasks. What if the downstream task requires a long context window then what can be the best set of eviction strategies and the corresponding expected win rates?
### Minor comments:
- "The resulting distribution is visualized in Figure As in Figure 3." can be rewritten as " Figure 3 shows the resulting distribution"
- A minor nit, the paper has too much forward referencing, which disturbs the flow of reading and attention, general recommendation in research papers is to avoid such referencing..!
- Better to define the new terms such as win-rate, KV cache budget, etc. when they were introduced for the first time. Similar applies to abbreviations when they are introduced first time, expand them, for the sack of saving readers time to search internet.

**Questions:**

Please refer to weaknesses section for questions.

## Post rebuttal comments

The responses and the detailed analysis in the Tables1,2 address my concerns.

However the authors seem to reserve one of the suggestions to the future works. Overall, very satisfied with the impressive work in the paper and raising the score to clear accept.

---

> ### Author Response · Authors · 2023-11-23
> **Response to Reviewer BiHE**
>
> **W1: What is the overhead of book-keeping to support this adaptive and diverse ability based on the type of the attention? What is the added computational complexity both asymptotically as well experimentally?**
>
> A1: Thanks for the valuable advice. We provide extra experimental results on book-keeping overhead in **Table 2 of General Response**. Please refer to the second part of General Response for a detailed time and memory analysis of profiling cost.
>
> In short, Table 2 shows the profiling time of the LLaMA65b in different generation length settings. We can observe that the profiling time only accounts for a very small percentage of the total generation duration, up to 0.35% in our tested cases. Also, the overhead decreases as the generation length increases, dropping to 0.07% when the generation length comes to 1024.
>
> In terms of extra memory usage, it’s mainly introduced by one of the compression strategies, C_frequent, which needs to store an extra cumulative sum of attention scores for each attention head.  To provide a detailed analysis, for each layer, the dimension of the KV cache is (batch_size, num_of_head, sequence_len, hidden_dimension), while the dimension of extra memory for the cumulative attention scores is (batch_size, num_of_head, sequence_len). Considering hidden_dimension=128 for all model sizes, the memory overhead is 1/128=0.78% compared to storing KV cache only, which is a negligible cost.
>
> In conclusion, the overhead introduced by the profiling step is nearly negligible in both time and memory, which confirms Fastgen’s potential for real-world deployment. We additionally provide end-to-end system latency improvement in Table 1. It shows that FastGen can achieve major speed-up in various generation settings. Please refer to the first part of General Response for more analysis.
>
> **W2: Table 3 shows an ablation on the policy order, why is this needed?  Is the policy fixed per layer and determined by the diagnosis step?**
>
> A2: The policy is determined in the diagnosis step, and it is fixed per head in each layer. As introduced in section 3.4 “Hybrid Policies”, we search for the optimal hybrid policy according to a predefined order. The order is greedily designed to prioritize cache policy with smaller memory costs, e.g., C_special. Once the optimal policy is determined, it will stay fixed in the generation process.
>
> In Table 3, the order ablation study aims to show that FastGen is agnostic to small changes in searching order. By shuffling the relative order of C_punct and C_local, we observe a different trade-off between KV cache compression and generation quality. Overall, our current order (as in Equation 2) achieves the highest win-rates.
>
> **W3: Another interesting exploration is long context tasks. In long context tasks, what can be the best set of eviction strategies and the corresponding expected win rates?**
>
> A3: Thanks for the suggestion, it points out a very promising area of extension for FastGen. Recent work such as StreamingLLM [1] and LM-Infinite [2] show that on long context tasks, preserving KV cache for special token and local contexts (C_special+C_local in our setting) could extend LLaMa-2 to 32k context length without significantly sacrificing performances. Although there is no clear experimental evidence on the optimal eviction ratio and corresponding win rates, their findings indicate that KV cache pruning could extend an LLM context length on-the-fly. It would be interesting to try out FastGen-style adaptive pruning on long context scenarios and measure performance and pruning ratio tradeoffs.
>
> [1] Xiao, Guangxuan, et al. "Efficient streaming language models with attention sinks." arXiv preprint arXiv:2309.17453 (2023)
> .
> [2] Han, Chi, et al. "Lm-infinite: Simple on-the-fly length generalization for large language models." arXiv preprint arXiv:2308.16137 (2023).
>
> We really appreciate the reviewer’s suggestions on our writing. We will reduce the forward referencing and improve the term definition in the future version.

---

> > ### Comment · Reviewer_BiHE · 2023-11-23
> >
> > Acknowledge the response from authors.

---

> ### Comment · Area_Chair_agrp · 2023-12-04
> **[Important] Detailed feedback required**
>
> Dear Reviewer BiHE,
>
> I noticed that there were no details accompanying this response. The detailed feedback is essential for the authors to understand the strengths and weaknesses of their work as perceived by the reviewers.
>
> Thank you,
>
> ICLR 2024 Area Chair

---

### Official Review · Reviewer_XU71 · 2023-11-01

**Soundness:** 3 good
**Presentation:** 4 excellent
**Contribution:** 3 good
**Rating:** 8
**Confidence:** 5

**Summary:**

This paper discussed how to apply adaptive KV cache compression to improve the system efficiency, which conducts profiling to discern the intrinsic structure of attention modules. The proposed method can be deployed without resource-intensive fine-tuning or re-training. Solid empirical study was conducted to verify the efficiency and effectiveness of the proposed method.

**Strengths:**

- This paper solves a critical research problem about efficient LLM inference with advanced algorithm design. The designed algorithm is straightforward and effective.

- The presentation of the technical discussion is accurate and well-organized.

- The organization of the evaluation sections is clear, and the presented results show the advance and efficiency of the proposed method.

**Weaknesses:**

- Based on my understanding, the proposed algorithm specializes in the most classic softmax-based attention. Is it possible to include a small section discussing the limitations of the proposed algorithm for more complicated attention mechanisms and some preliminary ideas about supporting those mechanisms in the future?

- Given the scale of the benchmarked model (llama-70B fp16 on A100-80G), I guess there is a missing detail about the parallel strategies applied in the experiments.

**Questions:**

Would it be possible to address the minor issues I listed in the weakness section?

---

> ### Author Response · Authors · 2023-11-23
>
> W1: The proposed algorithm specializes in classic softmax-based attention. Is it possible to include a discussion on more complicated attention mechanisms and some preliminary ideas about supporting those mechanisms?
>
> R1: Thanks for the suggestion. We are working on extending FastGen to other attention variations. One direct application is grouped query attention (GQA). In GQA, heads within each group share the same KV vectors. Instead of head-wise pruning, we could modify FastGen to perform group-wise pruning. Specifically, we could individually evaluate each query by calculating the recovery ratio of its attention map (Q*K), and then average all ratios within the same group, using the averaged ratio as the criteria to find the optimal strategy.
> We are not quite sure what specific "more complicated attention mechanisms" the reviewer has in mind, so it would be great if the reviewer could provide more references to them. We would be happy to include a discussion of these in the future version.
>
> W2: Given the scale of the benchmarked model (llama-70B fp16 on A100-80G), I guess there is a missing detail about the parallel strategies applied in the experiments.
>
> R2: We will add more details about our implementation in the future version. During inference, we perform model parallel by equally sharding the model weights to different GPUs within the same node. Attention heads are evenly distributed across GPU for parallel attention computation. During inference, the model_parallel_size is 8 for 70B, 4 for 30B, 2 for 13B, and 1 for 7B. During finetuning, we use 32 A-100 80G GPUs with model_parallel_size=4, data_parallel_size=8,  and batch_size_per_GPU=4 for all model sizes.

---

> > ### Comment · Reviewer_XU71 · 2023-12-04
> > **Thank you for your feedback**
> >
> > W.r.t. R1, in fact, I just wanted to refer to mechanisms such as GGA by "more complicated attention mechanisms".
> >
> > In general, thank you for your feedback! I do not have additional comments.

---

> ### Comment · Area_Chair_agrp · 2023-12-04
> **[Important] Response Required to Authors' Rebuttal**
>
> Dear Reviewer XU71,
>
> As we progress through the review process for ICLR 2024, I would like to remind you of the importance of the rebuttal phase. The authors have submitted their rebuttals, and it is now imperative for you to engage in this critical aspect of the review process.
>
> Please ensure that you read the authors' responses carefully and provide a thoughtful and constructive follow-up. Your feedback is not only essential for the decision-making process but also invaluable for the authors.
>
> Thank you,
>
> ICLR 2024 Area Chair

---

### Official Review · Reviewer_XPHq · 2023-11-01

**Soundness:** 3 good
**Presentation:** 3 good
**Contribution:** 3 good
**Rating:** 8
**Confidence:** 4

**Summary:**

This study introduces a lossy adaptive KV cache compression technique aimed at reducing the memory footprint of LLMs. The paper is guided by two key insights:

Different attention heads typically exhibit distinct structures.
These attention head structures remain relatively consistent during inference.
The paper profiles the prompt encoding phase to identify the intrinsic structures of various attention heads and uses these structures to determine the optimal compression policy. This policy, determined during the prompt encoding phase, is then applied uniformly throughout all token generation iterations. The compression policies are combinations of the following four basic ones: special, punct, local, and frequent.

The results demonstrate that this approach yields improved model quality compared to fixed KV compression methods, with KV cache budgets ranging from 30% to 100%. The ablation study further reveals that frequency- and special-token-based compression policies have the most significant impact on compression ratio and win rate.

**Strengths:**

- The paper introduces valuable insights drawn from LLMs: 1. Different structure in different attention 2. The same head structures persist. These insights are well-supported with empirical data and references to existing literature.

- The authors leverage these insights to come up with an effective compression method that adapts to the structure of each attention head. The results show consistent compression rate and model quality improvement over prior SoTA fixed compression mechanisms.

**Weaknesses:**

- The paper could benefit from presenting actual GPU inference performance results using FastGen and comparing them with other compression methods. Additionally, providing a runtime breakdown would offer more insights into the overhead caused by the profiling, compression, and decompression processes.
- It would be nice to look into the structure of KV in the multi-query attention design.

**Questions:**

-

---

> ### Author Response · Authors · 2023-11-23
> **Response to Reviewer XPHq**
>
> **W1: The paper could benefit from presenting actual GPU inference performance and comparing them with other compression methods.**
>
> R1: Thanks for the valuable advice. We provide extra experimental results in Tables 1 and 2 of **General Response** section. Please refer to the general response for a detailed analysis of actual end-to-end latency and profiling cost.
>
> In short, we can observe from Table 1 that FastGen achieves significant end-to-end speed-up across all the generation settings. For example, given batch_size = 1, FastGen’s relative speed-up rises from 16.04% to 55.0%, as the generation length grows from 512 to 16k. The phenomenon can also be observed in other batch settings. This analysis confirms that FastGen can achieve major speed-up in real development, especially in long-generation settings.
>
> Additionally, Table 2 shows the profiling time of the LLaMA65b in different generation length settings. We can observe that the profiling time only accounts for a very small percentage of the total generation duration, up to 0.35% in our tested cases. Also, the overhead decreases as the generation length increases, dropping to 0.07% when the generation length comes to 1024.
>
> In conclusion, the overhead introduced by the profiling step is nearly negligible, which confirms FastGen’s potential for real-world deployment.
>
> **W2: It would be nice to look into the structure of KV in the multi-query attention**
>
> R2: Thanks for the suggestions. Since multi-query attention essentially eliminates all KV heads to one, it already eliminates the memory cost and inference time to a large extent, leaving the benefit of pruning the KV cache marginal. Moreover, it could be non-trivial to find a universal pruning strategy for all query heads so we leave the exploration for future work. However, we agree that it would be interesting to look into this setting.

---

> ### Comment · Area_Chair_agrp · 2023-12-04
> **[Important] Response Required to Authors' Rebuttal**
>
> Dear Reviewer XPHq,
>
> As we progress through the review process for ICLR 2024, I would like to remind you of the importance of the rebuttal phase. The authors have submitted their rebuttals, and it is now imperative for you to engage in this critical aspect of the review process.
>
> Please ensure that you read the authors' responses carefully and provide a thoughtful and constructive follow-up. Your feedback is not only essential for the decision-making process but also invaluable for the authors.
>
> Thank you,
>
> ICLR 2024 Area Chair

---

### Author Response · Authors · 2023-11-23
**General response for all reviewers**

# Updated End-to-end Latency Improvement
To address reviewers’ concerns on the end-to-end speedup of FastGen, we implement a sparsity kernel for KV-cache pruning and present the end-to-end latency improvement in Table 1.

In the experiment, we record the total duration in seconds, measured from the start of prompt encoding, until the end of generation as the end-to-end latency.

For the Full-cache baseline, we use the widely used Hugging Face Accelerate (HF). For FastGen, we implemented a customized kernel to handle the KV cache pruning operation.  Specifically, we adapt the kernel from Deepspeed by adding the KV cache sparsity operation.
All methods are tested on the same Nvidia V100 GPUs.

**Table 1**: End-to-end latency comparison between Full-cache and FastGen on LLaMA7b. The column name "Settings" represents [Batch size, Prompt length, Generation length] tested. Each cell is the latency measured in seconds.

| Settings | [1,32,512] | [1,32,2048] | [1,32,8192] | [1,32,16384] | [2,512,32] | [2,512,512] | [2,4096,4096] | [8,512,512] | [8,4096,4096] | [16,512,512] |
| ------------------------ | ---------- | ----------- | ----------- | ------------ | ---------- | ----------- | ------------- | ----------- | ------------- | ------------ |
| Full-cache               | 13.35      | 57.37       | 299.00      | 799.14       | 1.12       | 19.16       | 167.64        | 1.97        | OOM           | OOM          |
| Fastgen (optimized ATTN) | 11.21      | 44.60       | 179.43      | 359.83       | 0.73       | 9.71        | 76.93         | 1.15        | 82.16         | OOM          |
| Speed-up(%)              | 16.03%     | 22.3%       | 40.0%       | 55.0%        | 34.8%      | 49.3%       | 54.1%         | 41.6%       | \-            | OOM          |

As shown in Table 1, we can observe that FastGen achieves significant end-to-end speed-up across all the generation settings. For the least significant case, Fastgen can have a decent 16.04% latency improvement over the full-cache baseline on a short generation length of 512. In the best cases, we can achieve up to 55.0% latency reduction with Fastgen at a generation length of 16k.

We can also observe that the relative speedup is greater with longer generation length. For example, given batch_size = 1, FastGen’s relative speed-up rises from 16.04% to 55.0%, as the generation length grows from 512 to 16k. The tendency can also be observed in other batch settings.

This analysis confirms that FastGen can achieve major speed-up in real development, especially in long generation settings. Meanwhile, the efficiency of the customized kernels can be further improved. We leave this unique research and engineering challenge to future works.

# Updated Analysis on the Profiling Cost (Time+Memory)
To better understand the overhead of the profiling step, we compare the profiling time with the total generation time across different generation lengths. We present the result in **Table 2**.

**Table 2**: Profiling time of LLaMA65b. The Overall Generation Duration is measured from the start of decoding to the end of the generation length. The Profiling Duration is measured from the start of the decoding until Fastgen finishes the policy search.
| Generation Length | Overall Generation Duration (s) | Profiling Duration (s) | Profiling/Overall (%) |
| ----------------- | ------------------------------- | ---------------------- | --------------------- |
| 128               | 30.98                           | 0.11                   | 0.35%                 |
| 256               | 50.10                           | 0.11                   | 0.21%                 |
| 512               | 94.98                           | 0.11                   | 0.12%                 |
| 1024              | 157.43                          | 0.11                   | 0.07%                 |

Table 2 shows the profiling time of the LLaMA65b in different generation length settings. We can observe that the profiling time only accounts for a very small percentage of the total generation duration, up to 0.35% in our tested cases. Also, the overhead decreases as the generation length increases, dropping to 0.07% when the generation length comes to 1024.

In terms of extra memory usage, it’s mainly introduced by one of the compression strategies, C_frequent, which needs to store an extra cumulative sum of attention scores for each attention head.  To provide a detailed analysis, for each layer, the dimension of the KV cache is (batch_size, num_of_head, sequence_len, hidden_dimension), while the dimension of extra memory for the cumulative attention scores is (batch_size, num_of_head, sequence_len). Considering hidden_dimension=128 for all model sizes, the memory overhead is 1/128=0.78% compared to storing KV cache only, which is a negligible cost.

In conclusion, the overhead introduced by the profiling step is nearly negligible in both time and memory, which confirms FastGen’s potential for real deployment.

---

### Public Comment · ~Arthfael_Yuuki2 · 2024-08-14
**Request for Code and Dataset Release for Awarded ICLR Paper**

I am writing to follow up on the promised release of the code and kernels associated with your paper presented at ICLR 2024. As of August 14, 2024—three months after the conference—the GitHub repository (https://github.com/machilusZ/FastGen) linked to your paper remains empty. This is concerning, particularly given the paper's significant impact and the recognition it received at ICLR.

It is important for the community that research is both transparent and reproducible. I urge you to make the research code and dataset publicly available as soon as possible. I am also bringing this to the attention of the ICLR program chair, as this issue affects the broader community's ability to build upon your work.

Thank you for your attention to this matter. I look forward to your prompt response.

---

> ### Public Comment · ~Yunan_Zhang1 · 2024-08-14
>
> Hi Arthfael, thanks for checking in! Sorry for the delay, I'm still working on the code release. Please understand I'm a full-time employee at Microsoft AI, where we have to wrap-up projects in very short time interval and my bandwidth is not controlled by myself. For now, there's very nice reproduction from the open-source community: https://github.com/AnswerDotAI/cold-compress/tree/main. You can use --cache_strategy hybrid to try the FastGen algorithm.
> Meanwhile, you can also check MInference(also from Microsoft): https://github.com/microsoft/MInference/tree/main, which is a very optimized codebase, and can be viewed as a close implementation to the FastGen algorithm. Core part is the same, taking the last q at prefilling stage with K to do a profiling on attention mat, then decide for each attention head, whether to adopt topk, first/last token(aka special tokens), block sparse sparsity pattern. There's a change in threshold choice by constraining the total FLOPs for each dataset, which I think they design to overcome time-space tradeoff.

---

> > ### Public Comment · ~Arthfael_Yuuki2 · 2024-08-15
> > **Follow-up comment on Code**
> >
> > Hi Yunan,
> >
> > Thank you for your response and for sharing the alternative resources. I understand the challenges of balancing full-time work with research commitments, and I appreciate the effort you're putting in.
> >
> > However, I must emphasize the importance of fulfilling the promise in your paper's abstract to release the official code and CUDA kernels. As stated in the paper, **"We will release our code and the compatible CUDA kernel for reproducibility."** This commitment is crucial for ensuring the community can accurately reproduce your results and build upon your work. While third-party implementations can be helpful, they do not substitute for an official release, especially when verifying the exact methodologies and numbers presented in your paper.
> >
> > Given that this paper was accepted in January, the community has been waiting for the official code for several months. Delays in releasing this code can damage the progress of subsequent research and diminish the paper's overall impact. I believe this open-sourcing promise was one of the factors in the paper's acceptance and recognition at ICLR, and it is essential to uphold these standards.
> >
> > I urge you to prioritize the release of the official code and dataset as soon as possible.
> >
> > Thank you for your attention to this matter. I look forward to the official release.

---

### Public Comment · ~Anastasiia_Filippova2 · 2024-11-27
**Confusing references in the related work**

Dear Authors,

Thank you for the interesting and thought-provoking paper. I have a question regarding the statement in the related work section:
"Meanwhile, many efforts have been made to explore the possibility of compressing the hidden state of tokens rather than explicitly reducing the sequence length (Guan et al., 2022; Sun et al., 2022; Zhou et al., 2020)."

From my understanding, the cited works (e.g., Transkimmer by Guan et al., 2022, BERT Loses Patience by Zhou et al., 2020 and Sun et al., 2022) are primarily about early exiting or adaptive inference strategies rather than directly compressing hidden states. Could you clarify how these works relate to hidden state compression in your context, or if I might be missing a connection here?

Thank you in advance for your insights!

---

> ### Public Comment · ~Suyu_Ge1 · 2024-12-02
> **Response to related work**
>
> Hi Anastasiia,
>
> Thank you for your interest in our paper! The cited works mainly aim to reduce the size of hidden states of encoder-based transformers. For example, the first layer and last layer of transkimmer remain the full sequence length. In the middle layers, the evicted position of transkimmer is not always consistent. The 5th token may be retained in the 3rd layer but evicted in the 4th layer. On the contrary, we employ consistent eviction from the first to the last layer.
>
> Feel free to ask if you have any other questions!

---

### Meta-Review · Area_Chair_agrp · 2023-12-06

**Metareview:**

The paper under consideration introduces innovative insights into Large Language Models (LLMs) and proposes an effective compression method for attention heads in these models. The average rating from reviewers is an 8, which indicates a general consensus towards the high quality and relevance of the work. The strengths of the paper, as highlighted by multiple reviewers, include its valuable insights into LLMs, effective compression methods, clarity in presentation and organization, and the novel concept of an adaptive KV cache.

Given the high ratings and the paper's contributions to the field, I am inclined to recommend acceptance. The paper addresses a research problem in efficient LLM inference with a well-designed algorithm and a clear presentation of its technical aspects and evaluation. The insights it offers are substantial, particularly in the context of model-specific attribute alignment and memory footprint reduction during generative inferences.

**Justification For Why Not Higher Score:**

N/A

**Justification For Why Not Lower Score:**

The average of this paper is eight, and received a high consensus of acceptance from the reviewers.

---

### Decision · Program_Chairs · 2024-01-16

Accept (oral)